

# Biomass allocation, carbon content change and carbon stock distribution of Scots pine (*Pinus sylvestris* var. *mongholica*) plantation forests at different stand ages and densities in the sandy area of western Liaoning Province, China

Xin Ai[1,2], Xiangyu Yang[1,2], Zhaowei Zhang[1,2], Hao Chen[1,2], Wenhui Tang[1,2], Qingyu Wang[1], Yutao Wang[1,2] and Ping Liu[1,2,3]

[1] College of Forestry, Shenyang Agricultural University, Shenyang, Liaoning Province, China
[2] Key Laboratory for Silviculture of Liaoning Province, Shenyang Agricultural University, Shenyang, Liaoning Province, China
[3] Engineering Technology Research Center of Chinese Pine of National Forestry and Grassland, Shenyang, Liaoning Province, China

Corresponding authors
Yutao Wang, ytw730@syau.edu.cn
Ping Liu, lp_79@syau.edu.cn

## ABSTRACT

Scots pine (*Pinus sylvestris* var. *mongholica*) is one of the main afforestation species in the southeastern edge of the Horqin Sandy Land, which not only effectively prevents the expansion of the sandland, but also serves as an important carbon reservoir. Uncovering the biomass allocation, carbon content changes and carbon stock distribution among organs of Scots pine at different ages and densities can provide a theoretical basis for rational afforestation and management in the western Liaoning sandy area. In this study, the biomass and carbon content of four organs, namely, trunk, branch, leaf and root, were measured at different age classes (young stage, half-mature stage, near-mature stage, mature stage and over-mature stage forests) and densities, and the carbon stock of Scots pine plantations in the western Liaoning sandy area was estimated. The results showed that the biomass of all organs except leaves increased with the increase of stand age, but the rate of increase of each organ was not consistent. To resist wind and sand, the biomass was preferentially allocated to the trunk and roots, which was in line with the theory of allometry and optimal allocation. The carbon content of each organ of Scots pine increases and then decreases with the rise of forest age classes, and the root carbon content is the lowest in five forest ages, and the plant carbon is mainly stored in the aboveground part. The biomass of each organ in both near mature and mature forests increased with the decrease in density. Still, the root carbon content decreased with the decrease of density, and the PCA analysis showed that near mature and mature forests had better carbon sequestration capacity in low density. The carbon stock of Scots pine plantation forests in the sandy area of western Liaoning was mainly concentrated in Fuxin and Chaoyang cities, and the lowest carbon stock was found in Jinzhou. The age and density of the forest stand are important factors affecting the biomass and carbon content of Scots pine, therefore, when operating Scots pine plantation forests in the

sandy areas of western Liaoning, different stand densities should be retained at different age stages, so that their biomass and carbon content can be sufficiently accumulated and distributed to improve the local environment.

## INTRODUCTION

Land desertification has long plagued countries worldwide, and China put forward the goal of preventing and treating sand in the Outline of the Fourteenth Five-Year Plan and Vision 2035 for the National Economic and Social Development of the People's Republic of China (*The People's Government of Fujian Province, 2021*). To accomplish the goal of preventing and treating sand, in June 2023, Liaoning Province formulated and issued the Liaoning Province Horqin Sand Land Annihilation and Desertification Comprehensive Prevention and Control Action Programme (2023–2030), which divides the action programme into four management zones, and the western Liaoning sandy is located in one of them, the Horqin sand land annihilation and attack zone. The cities and counties of western Liaoning have serious land sandification, which makes the goal of sand prevention and control a challenge, while the high survival rate of Scots pine (*Pinus sylvestris* var. *mongholica*) plantation forests in semi-arid sandy areas brings hope for the completion of the sandland annihilation war and desertification control. The United Nations Framework Convention on Climate Change (UNFCCC) was adopted by the United Nations General Assembly in 1992, and it entered into force for China in 1994 (*Kuyper, Schroeder & Linner, 2018*; *Nasiritousi et al., 2024*), we should focus more on estimating forest carbon to better understand how forest carbon stocks reduce $CO_2$ emissions. With increasing global warming and rapid ecological degradation, carbon stocks in arid regions are gradually decreasing (*Zhu et al., 2021*), yet the biomass and carbon stocks of sandy Scots pine have not been sufficiently studied (*Li et al., 2018*; *Li et al., 2013*).

Forest biomass is the result of the long-term production and metabolism of forest ecosystems (*Mao et al., 2020*), and forest biomass is the most basic feature of the structure and function of forest ecosystems. Carbon content is the content of organic carbon in living organisms, and forest carbon stock can be indirectly projected through biomass and carbon content, which reflect the potential and ability of forest communities to utilise natural resources, and are not only important indicators of forest productivity, but also the basis for studying the material cycle of forest ecosystems (*Chen et al., 2018*; *Ruiz-Benito et al., 2014*; *Sullivan et al., 2020*). The status and changes in forest biomass carbon stocks are essential for the development of sound national policies on the management and care of forest resources at large spatial scales (*McKinley et al., 2011*). Measurement of forest biomass and carbon content is important for in-depth studies of biogeochemical cycling and carbon sink functions of forest ecosystems, but also for the conservation of terrestrial biodiversity on a global scale (*Bai et al., 2023*; *Jung et al., 2021*; *Liu et al., 2023*; *Ruehr et al.,*

*2023*). In their assessment and prediction of carbon stocks in Chinese plantation forests from 2023 to 2060, *Chen et al. (2024)* showed that the highest carbon stocks are expected in northeastern China by 2060. Therefore, the estimation of carbon stocks in plantation forests of various tree species in Northeast China is imperative. There are various methods for estimating tree biomass, and aboveground biomass is generally estimated based on physiological process method, based on forest inventory method, forest harvesting method, allometry equations, remotely sensed data, and spectral analyses (*Guo et al., 2009*; *Ishihara et al., 2015*; *Liu et al., 2019*; *Liu et al., 2024*; *Xiang et al., 2011*; *Yue et al., 2023*). Estimation of biomass in the below-ground portion is generally done by harvesting method, root drilling method, internal growth method, micro-root window method, and isometric scaling using allometry equations (*Wang et al., 2017*; *Roberti et al., 2018*; *Wei, Xiaochan & Fengjie, 2019*; *Ditsouga et al., 2024*). The main methods for estimating forest carbon stocks are inversion through statistical analyses of models as well as remote sensing (*Huang et al., 2019*; *Weiskittel et al., 2015*), while most of these methods involve tree biomass equations, tree allometry equations tend to apply only to forest stands under specific conditions (*Zhou et al., 2021a*; *Zhou et al., 2021b*). Another commonly used method is indirect calculation through biomass and carbon content (*Ge et al., 2013*; *Sun & Liu, 2019*; *Zhao et al., 2018*).

While the biomass of forest ecosystems is influenced by a combination of factors such as species composition, stand age, climate, site conditions and anthropogenic disturbances, there has been less research on the factors influencing the carbon content of trees. The current research on tree biomass and carbon content is far from adequate, both in terms of stand types and eco-geographical region types, which inevitably leads to obvious uncertainties in the assessment of forest productivity and carbon sink functions in different regions and scales (*Qi et al., 2019*). The coordinated development between different organs of an individual plant is a life-history response to plant growth, and the distribution of energy and matter between organs is influenced by a variety of conditions (*Byambadorj et al., 2024*; *Ciais et al., 2005*; *Poorter & Sack, 2012*), such as age, external temperature, moisture, light and environmental conditions such as planting density. The theory of allometry growth and optimal allocation in plants has been accepted by most scholars, which suggests that the allocation of plant biomass is most fundamentally governed by the size of the individual, and that the accumulation of biomass in different organs shows an allometry growth relationship (*Gargaglione, Peri & Rubio, 2009*; *Puglielli et al., 2021*). The optimal allocation theory, on the other hand, suggests that when a plant encounters a certain resource limitation, the plant will preferentially allocate metabolites to organs that have access to the limiting resource (*McCarthy & Enquist, 2007*). However, although this theory has all been validated in plants from different ecosystems, the effects of multiple global change factors, both individual and combined, on plant organ biomass allocation strategies have yet to be tested (*Peng et al., 2021*).

With the characteristics of cold-resistant, drought-resistant and barrenness-resistant, Scots pine is one of the main afforestation species in the sandy land of western Liaoning and plays a key role in stopping the expansion of the Horqin sandy land. Currently, studies on biomass allocation among organs in different densities of Scots pine mainly focus on the above-ground portion of mature forests (*Siqing et al., 2022*; *Wertz et al., 2020*), whereas

studies on carbon content changes mainly focus on young forests or soils (*Baumann et al., 2006*; *Zhang et al., 2022*), which is a lack of information on the biomass allocation of Scots pine. There is a lack of studies on the biomass and carbon content of Scots pine throughout its life cycle, and no study has estimated the carbon stock of Scots pine in the sandy areas of western Liaoning. In this study we hypothesized that (I) the biomass allocation of each organ in the plantation forest of Scots pine follows the theory of allometry and optimal allocation; (II) the carbon content of each organ of Scots pine varies in different forest ages and densities; and (III) the capacity of carbon sequestration of Scots pine varies in different densities. In this way, the biomass distribution, carbon content change and carbon stock distribution among organs of Scots pine at different ages and densities can be clarified, and provide theoretical basis for the rational afforestation and management of sandy land in western Liaoning.

## MATERIAL AND METHODS

### Study site

The study area is located in Zhanggutai Town Forest, Zhangwu County, Liaoning Province, with geographic coordinates between 42°39′N and 42°43′N, 122°28′E and 122°34′E, and an average elevation of 218 m. It is situated at the southeastern edge of the Horqin sandy land, with the terrain of north high and south low, east and west hilly, sandy wasteland in the northern part of the area, and plain in the middle and southern part of the area, which is subject to severe wind and sand erosion. It belongs to the temperate continental monsoon climate, 2023 average annual temperature of 16 °C, the highest average daily temperature of 36 °C, the lowest average daily temperature was −25 °C, the hottest month appeared in July, the average temperature of 30 °C, the coldest month appeared in January, the average temperature of −15 °C, 2023 annual precipitation of 486 mm, the annual maximum monthly precipitation of 238.2 mm (July), the annual minimum monthly precipitation of 0 mm (March). The soil is dominated by wind-sand soil, which is a typical semi-arid wind-sand land (Fig. 1). In order to stop the expansion of the Horqin Sand Land, one of the main local sand fixation and afforestation species is the Scots pine (*Pinus sylvestris* var. *mongholica*).

### Data collection

#### Selection of standard land and standard wood

From May to October 2023 in the forest farm of Zhanggutai Town, Zhangwu County, Liaoning Province, according to the National Forestry Administration (2017; Supplementary files), standard plots (20 m×30 m) were set up within the Scots pine plantation forests with basically the same stand conditions, three plots of young stage (YS), three plots of half-mature stage (HMS), nine plots of near-mature stage (NMS), nine mature stage (MS) and three over-mature stage forests (OMS), totaling 27 standard plots. According to the Regulations for Forest Tending (2015), young and middle forest age is the main period for adjusting the density, while near-mature and mature forests are the stable period for retaining the density, to explore the appropriate retention density of Scots pine, high density (HD), medium density (MD) and low density (LD) were set up in the

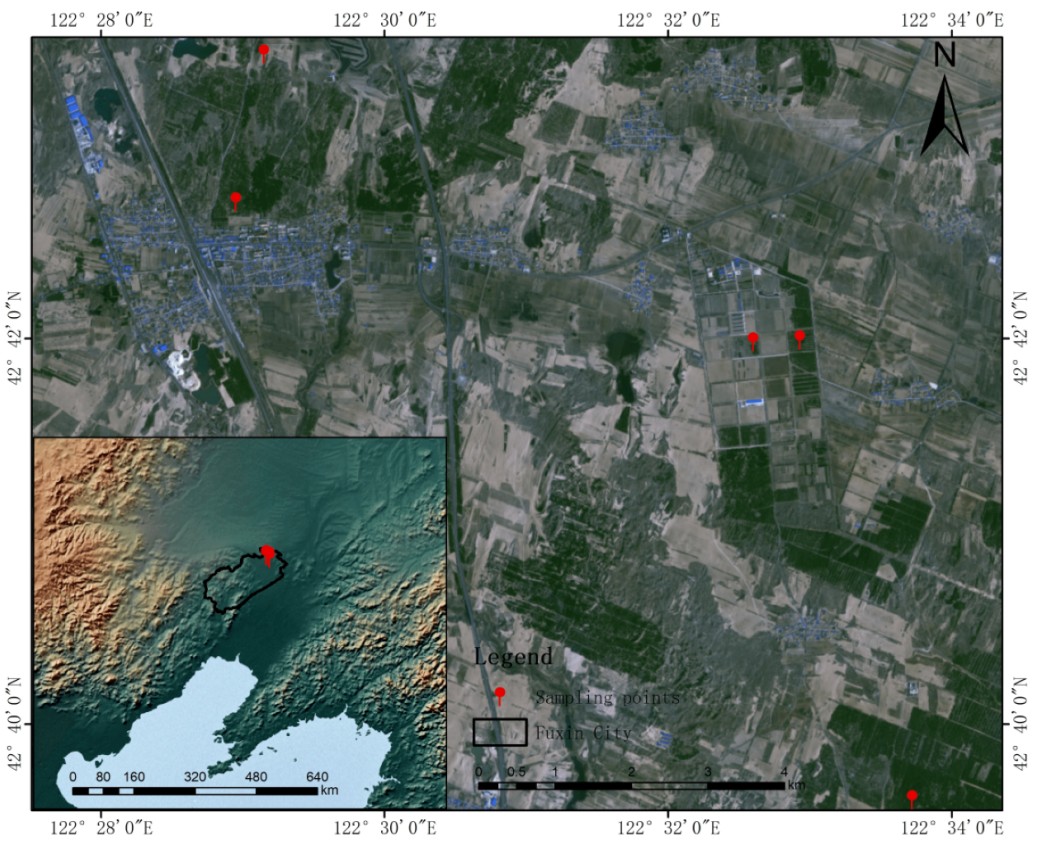

**Figure 1** **Schematic map of sampling locations.** Specific sampling points are marked in red, and maps are made from natural earth, derived from free vector and raster data @ naturalearthdata.com.

near-mature forests and mature forests, respectively three standard plots. The diameter at breast height (DBH), tree height, height under the branch, and crown spread of each Scots pine in the sample plots were measured using a diameter at breast height ruler, Blume-Leiss height gauge, and a tape measure, respectively (Table 1). Then, two Scots pines that were similar to the average diameter at breast height and tree height with straight trunks, free of pests and diseases, without scars, and with good growth were selected as standard trees in each standard plot, totaling 54 standard trees.

### Biomass measurement

*Measurement of trunk biomass.* The wood density method was used to calculate the biomass of the standard wood trunk (*Raden et al., 2020*). A growth cone with an inner diameter of 5.15 mm was used to drill the core of the standard wood in the north-south direction at the diameter of the chest (1.3 m), and the length of the core was measured and recorded with a ruler, and then brought back to the laboratory to be killed in an oven at 105 °C for 30 min and then dried at a constant temperature of 80 °C to a constant weight. Two cores were drilled from each standard tree, totaling 108 cores. Trunk biomass was calculated

**Table 1  Basic information of the sample plots of different forest ages of Scots pine.**

| Age class | Stand age (a) | Mean DBH (cm) | Mean tree height (m) | Mean branch height (m) | Stand density (tree· hm$^{-2}$) |
|---|---|---|---|---|---|
| YS | 9 | 4.55 ± 0.04 | 2.43 ± 0.10 | 0.33 ± 0.03 | 528 ± 23 |
| HMS | 22 | 11.89 ± 0.68 | 5.34 ± 0.65 | 0.76 ± 0.09 | 605 ± 29 |
| NMS | 33 | 16.38 ± 0.95 | 7.90 ± 0.25 | 3.67 ± 0.31 | 1178 ± 91 |
| NMS | 33 | 17.51 ± 0.50 | 9.38 ± 0.41 | 5.04 ± 0.47 | 772 ± 57 |
| NMS | 33 | 18.30 ± 1.05 | 7.86 ± 0.15 | 3.29 ± 0.34 | 506 ± 56 |
| MS | 41 | 17.92 ± 0.82 | 11.31 ± 1.10 | 6.79 ± 1.10 | 1,094 ± 122 |
| MS | 41 | 20.62 ± 1.03 | 12.00 ± 0.78 | 6.78 ± 0.86 | 650 ± 50 |
| MS | 41 | 25.16 ± 0.76 | 10.58 ± 0.33 | 3.93 ± 0.27 | 322 ± 18 |
| OMS | 61 | 28.63 ± 4.14 | 13.05 ± 0.67 | 6.70 ± 0.38 | 244 ± 23 |

**Notes.**

DBH stands for diameter at breast height, YS, HMS, NMS, MS and OMS denote young, half-mature, near-mature, mature and over-mature forests, respectively.

using the following formula:

$$TB = V_1 \times B. \tag{1}$$

In the formula, TB is the trunk biomass, $V_1$ is the standard wood volume (obtained through the binary wood volume table of Scots pine), and $B$ is the trunk basic density. The basic density of the trunk is approximated by the basic density of the wood core instead of the basic density $B = M_1/V_2$, $M_1$ is the dry weight of the wood core, and $V_2$ is the volume of the wood core.

*Determination of branch and leaf biomass.* The branch and leaf biomass of standard wood was determined by stratified sampling method (*Wang, 2022*). The canopy of standard trees was divided into upper, middle and lower layers, the number of branches in each layer was recorded, two standard branches were selected in each layer, and a total of six standard branches were intercepted manually with high pruning shears or saws, and the weight of each standard branch was weighed and recorded by electronic weighing, to reduce the error caused by the loss of moisture of needles in the process of needle picking, the intercepted standard branches were weighed and recorded with leaves, and then the needles and leaves were weighed again after removing them. Fresh weight of branches, respectively, selected sample branches, and sample leaves weighed fresh weight and brought back to the laboratory at 105 °C after killing 30 min, 80 °C drying to constant weight. Each standard wood to take six standard branches, a total of 324 standard branches. The biomass of the branches was calculated using the following formula:

$$BB_1 = \frac{M_3}{2} \times N_1 \times P_1. \tag{2}$$

In the formula, $BB_1$ is the branch biomass of a layer in the upper, middle and lower canopy, $M_3$ is the fresh weight of two standard branches (defoliated) in the corresponding layer of the canopy, $N_1$ is the number of branches in the corresponding layer of the canopy,

$P_1$ is the water content of the standard branches in the corresponding layer of the canopy (dry weight of the sample branch compared with the fresh weight of the sample branch), and the biomass of a single branch $BB$ is equal to the sum of the biomass of the branches of the three layers of the canopy. Six conifer samples were taken from each standard tree, totaling 324 sample leaves. Leaf biomass was calculated using the formula:

$$LB_1 = (M_2 - M_3) \times N_1 \times P_2. \tag{3}$$

Where $LB_1$ is the leaf biomass of a layer of the canopy, $M_2$ is the fresh weight of a standard branch (with leaves) of the corresponding layer of the canopy, $P_2$ is the water content of the leaves of the corresponding layer of the canopy, and the leaf biomass of a single plant, $LB$, is equal to the sum of the biomass of the leaves of the three layers of the canopy, respectively.

*Root collection and biomass determination.* The roots of the underground part of the standard wood were collected by soil auger sampling (*Li et al., 2017*), 10 points were determined along the S-shape in the standard land with the middle position of the row spacing of the camphor Scots pine plantation forest as the starting point, and the rooted soil cores were drilled with root augers with an inner diameter of eight cm at these 10 points in three layers (0–20 cm, 20–40 cm, and 40–60 cm). Roots with rooted soil cores were sieved through a 20-mesh mesh sieve to remove the soil and then washed and shade-dried, and the shade-dried roots were placed in an oven, killed at 105 °C for 30 min and then dried at 80 °C to a constant weight, and weighed using an electronic balance (accurate to 0.001 g). Ten sample roots were taken from each sample plot, totaling 270 sample roots. Root biomass was calculated according to the following formula:

$$RB = \frac{108m}{P\pi r^2}. \tag{4}$$

Where $RB$ is the root biomass of a single plant (g), $m$ is the average root dry weight per soil core (g), $r$ is the radius of the root auger (cm), and $P$ denotes the stand density of the sample plot at each age class (hm$^{-2}$).

*Determination of monocot biomass.* Scots pine single plant biomass $LTB$ is the sum of biomass of each organ. The formula is as follows:

$$LTB = TB + BB + LB + RB. \tag{5}$$

### Carbon content determination

Plant samples' whole carbon (C) content was determined by the volumetric method using potassium dichromate (*Cha, 2017*). After mixing the samples of the same organ under the same forest age and density of Scots pine, crushing them with a shredder and then passing them through a 60-mesh mesh sieve, then passing them through a 100-mesh mesh sieve, weighing 0.025 g of samples with an electronic balance (six copies of samples of each organ), adding five ml of $K_2Cr_2O_7$ solution at one mol/L, then adding 10 ml of concentrated

sulfuric acid, and then adding 50 ml of distilled water after cooling and adding three drops of 1,10-Phenanthroline hydrate and titrating with 0.3 mol/L of $Fe_2SO_4$ solution. Six groups of blank control were set to take the average value. The following formula was used to calculate the total carbon content of the samples:

$$C = [0.3 \times (v_0 - v) \times 0.003 \times 1.33]/0.025 \times 1000. \tag{6}$$

Where $C$ is the sample carbon content (g/kg), $v_0$ is the volume of $Fe_2SO_4$ standard solution consumed for blank test determination (ml), and $v$ is the volume of $Fe_2SO_4$ standard solution consumed for sample determination (ml). The carbon content $LTC$ of Scots pine monoculture is then:

$$LTC = (TC \times TB + BC \times BB + LC \times LB + RC \times RB)/ITB. \tag{7}$$

where $LTC$ is the whole carbon content of a single plant, $TC$, $BC$, $LC$ and $RC$ are the whole carbon content of the trunk, branches, leaves and roots, respectively, and $TB$, $BB$, $LB$ and $RB$ are the biomass of the trunk, branches, leaves and roots, respectively.

### Estimation of carbon stocks in Scots pine plantation forests in the sandy land of west Liaoning Province, China

Based on the forest resources inventory data of Liaoning Province and the Sixth Land Desertification and Sand Degradation Monitoring in China, a total of 851 Scots pine plantation forests (2023) located in the sandy areas of western Liaoning were screened out from them and divided into corresponding forest ages according to the national forestry industry standards and counted the number of Scots pine monocultures in each forest age. The product of biomass and carbon content of whole plants is calculated according to Eqs. (5) and (7) are the carbon stock of whole plants, and the formula is:

$$LCS = LTB \times LTC \tag{8}$$

where $LCS$ is the carbon stock of a single plant, and this was used as the basis for estimating the carbon stock of each forest age in the Liaoning Sand.

## Data analysis

Statistical analyses were performed using SPSS 25.0 (IBM Corp., Armonk, NY, USA) software. One-way ANOVA and Duncan's method of ANOVA and multiple comparisons ($\alpha = 0.05$) were used to analyze the differences in organ biomass and C content of Scots pine at different age stages and different densities and also to analyze the differences in organ biomass and C content of Scots pine monocultures at different age stages and different densities. A two-way ANOVA was used to analyze the interaction effects of stand age and density on organ biomass and C content of Scots pine. The carbon sequestration capacity of Scots pine at different densities was comprehensively evaluated by principal component analysis (PCA), and the carbon stock of Scots pine in the sandy area of western Liaoning was classified into five levels by the natural discontinuity classification (NBC) method. Relevant graphs were plotted using Excel 2016 (Microsoft Office, Redmond, WA, USA), Origin 2023 (OriginLab, Northampton, MA, USA), and ArcMap10.2 (ArcGIS, Redlands, CA, USA), and graphs of the data are mean $\pm$ standard deviation.

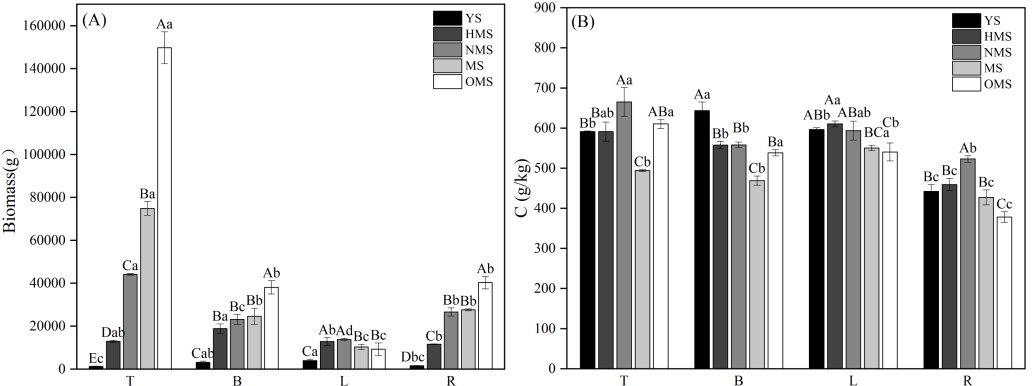

**Figure 2** **Biomass allocation (A) and carbon content changes (B) in various organs of Scots pine at different forest ages.** YS, HMS, NMS, MS and OMS denote young, half-mature, near-mature, mature and over mature forests, respectively; T, B, L and R denote trunks, branches, leaves and roots, respectively; different capital letters denote significant differences between forest ages, and different lowercase letters denote significant differences between organs ($P < 0.05$).

## RESULTS

### Effect of forest age on biomass and carbon content of different organs of Scots pine

The biomass and carbon content of each Scots pine organ at different forest ages showed that for biomass, except for L, T, B and R biomass increased with the increase of forest age. Among them, T biomass differed significantly ($P < 0.05$) among the five forest ages, with the lowest being 1,291.404 g in the YS and the highest being 149,787.264 g in the OMS; B and R biomass differed significantly ($P < 0.05$) among the forest ages except for the NMS and MS. In the YS, L biomass was the highest at 3,962.627 g, followed by B3, 100.223 g and R1, 592.859 g, and T was the lowest at 1,291.404 g. In the HMS, there was little difference in the biomass of individual organs (Fig. 2A). The pattern of biomass of each organ was the same in NMS, MS and OMS, which all showed that T was the highest, followed by R and B, and L was the lowest. The carbon content of each organ was lower in MS and OMS, and higher in YS, HMS and NMS. The L carbon content was high in all five forest ages, and the R carbon content was the lowest in all five forest ages (Fig. 2B).

The results of the biomass and carbon content of whole plants of Scots pine in different forest ages showed that the biomass of whole plants of Scots pine increased continuously with the increase of forest age, from 9,911.113 g in YS to 241,287.570 g in OMS. The difference between different forest ages was significant (P < 0.05) (Fig. 3A). The carbon content of single plant of Scots pine was the highest in the NMS, 585.134 g/kg, which was not significantly different from that of YS and HMS ($P > 0.05$), but considerably higher than that of OMS and MS (P < 0.05), and the lowest in mature forests, 485.423 g/kg (Fig. 3B). The results of the biomass proportion of each organ in different forest ages of Scots pine showed that the proportion of TB of Scots pine increased with the increase of forest age. In contrast, the proportion of BB and LB decreased with the increase of forest age,

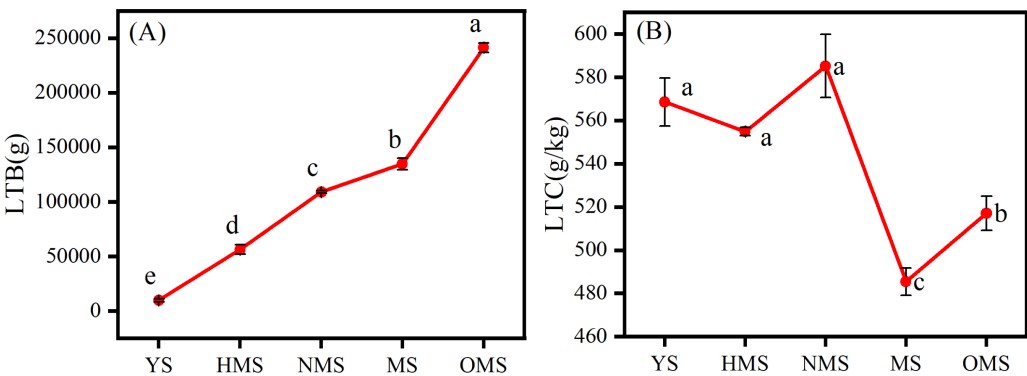

**Figure 3** **Distribution of whole plant biomass (A) and carbon content changes (B) of Scots pine at different forest ages.** LTB denotes whole plant biomass, LTC denotes whole plant carbon content, YS, HMS, NMS, MS and OMS denote young, half-mature, near-mature, mature and over-mature forests, respectively, and different lowercase letters denote significant difference ($P < 0.05$).

and the proportion of RB increased from 17% in YS to 26% in NMS and then decreased to 16% in OMS, with small fluctuation of the proportion (Fig. 4).

## Effect of density on biomass and carbon content of different organs of Scots pine

Note: HD, MD and LD denote high, medium and low densities, respectively; T, B, L and R denote trunk, branch, leaf and root, respectively; different capital letters denote significant differences between densities, and different lowercase letters denote significant differences between organs ($P < 0.05$).

The results of biomass and carbon content of each organ in the near-mature Scots pine at different densities showed that the biomass of each organ of Scots pine increased with the decrease of density, among which, the biomass of T and L was similar at medium and low densities, and significantly higher than that at high density ($P < 0.05$); the biomass of B and L was not significantly different at the three densities ($P > 0.05$); the biomass of R was the highest at the low density of 26,630.942 g and significantly higher than that at the other two densities ($P < 0.05$), and the lowest at high density was 14,164.319 g (Fig. 5A). 26,630.942 g, which was significantly higher than the other two densities ($P < 0.05$), and the lowest at high density at 14,164.319 g (Fig. 5A). The carbon content of all organs except R increased with decreasing density, among which, T and L did not differ significantly among the three densities, B was significantly higher ($P < 0.05$) than the high density at medium and low densities, and the carbon content of R did not differ significantly ($P > 0.05$) among the three densities (Fig. 5B).

Biomass of all organs in mature Scots pine increased with decreasing density (Fig. 6A). Carbon content of all four organs in mature Scots pine was highest in high density, B and L carbon content did not differ much between medium and low densities, while R was highest in high density at 580.944 g/kg, which was significantly higher than medium and low densities ($P < 0.05$) (Fig. 6B).

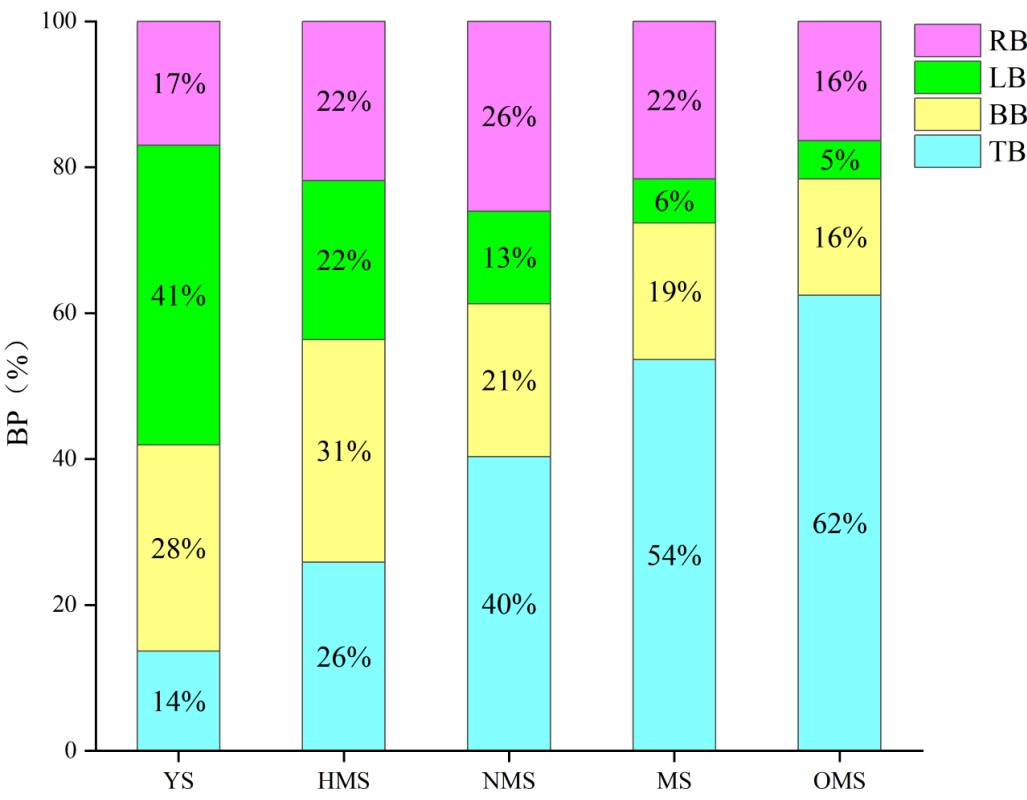

**Figure 4 Biomass proportion of each organ in different forest ages of Scots pine.** BP indicates biomass proportion, YS, HMS, NMS, MS and OMS denote young, half-mature, near-mature, mature and over-mature forests, respectively, and TB, BB, LB and RB denote trunk biomass, branch biomass, leaf biomass and root biomass, respectively.

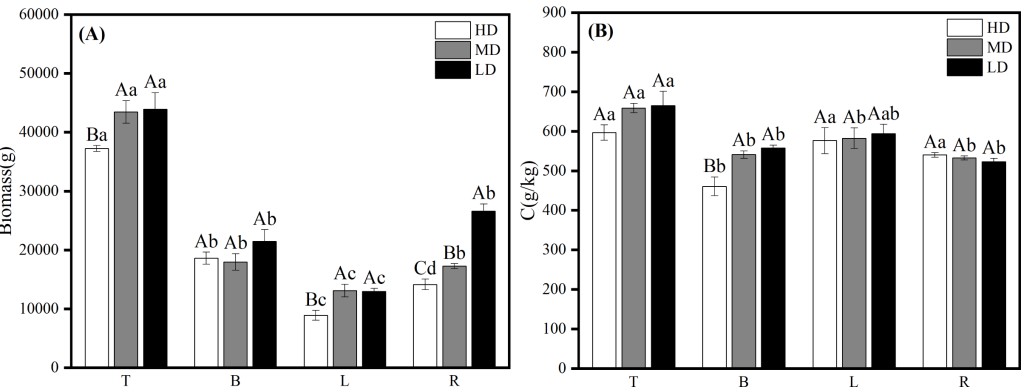

**Figure 5 Biomass distribution (A) and carbon content changes (B) of each organ in different densities of near-mature Scots pine.** HD, MD and LD denote high, medium and low densities, respectively; T, B, L and R denote trunk, branch, leaf and root, respectively; different capital letters denote significant differences between densities, and different lowercase letters denote significant differences between organs ($P < 0.05$).

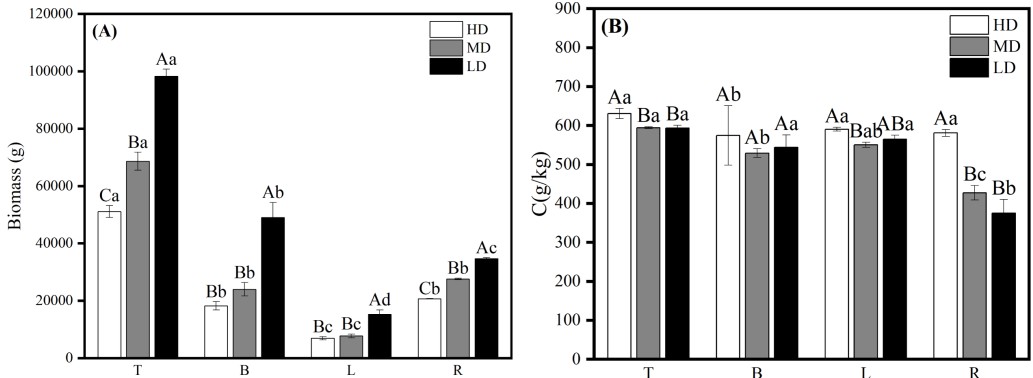

**Figure 6 Biomass allocation (A) and carbon content changes (B) of each organ in different densities of mature Scots pine.** HD, MD and LD denote high, medium and low densities, respectively; T, B, L and R denote trunk, branch, leaf and root, respectively; different capital letters denote significant differences between densities, and different lowercase letters denote significant differences between organs ($P < 0.05$).

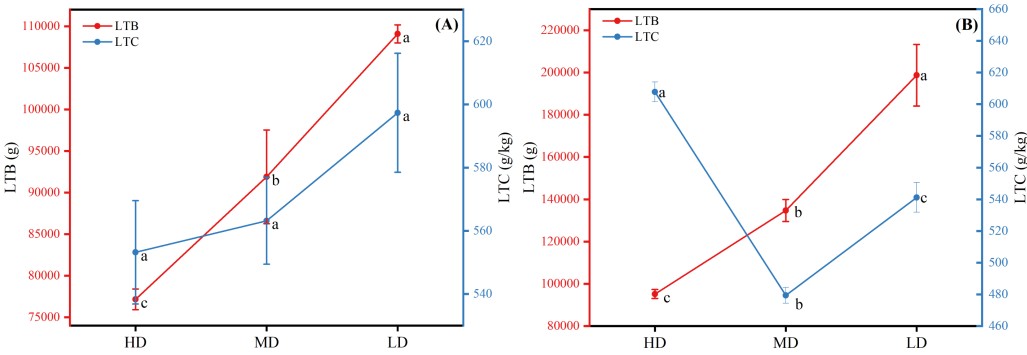

**Figure 7 Whole-plant biomass and full carbon content of near-mature forest (A) and mature forest (B) of Scots pine.** HD, MD and LD denote high density, medium density and low density, LTB and LTC denote whole-plant biomass and carbon content, respectively, and different lowercase letters denote significant differences ($P < 0.05$).

The biomass and carbon content of a single plant in the near-mature Scots pine both increased with the decrease of density, the biomass was 198,810.463 g in low density and 95,218.364 g in high density, and the difference in carbon content was not significant among the three densities ($P > 0.05$) (Fig. 7A). The biomass of single plant in mature Scots pine also increased with decreasing density, while the carbon content was the highest at high density, 607.774 g/kg, and the lowest at medium density, 479.450 g/kg (Fig. 7B).

## Comprehensive evaluation of carbon sequestration capacity of Scots pine at different densities in the sandy areas of west Liaoning Province

At the same age, plants with high biomass and carbon content imply better carbon sequestration capacity. Therefore, in this study, the carbon sequestration capacity of Scots pine of different forest ages was comprehensively evaluated from the two core dimensions of biomass and carbon content using principal component analysis (PCA). The PCA showed

**Table 2** Loadings and explained variance of Scots pine biomass and C content in PCA analyses for different densities of near-mature and mature forests.

| Index | Near-mature forests | | | Mature forests | |
|---|---|---|---|---|---|
| | PC1 | PC2 | PC3 | PC1 | PC2 |
| TB | 0.443 | −0.184 | −0.200 | 0.452 | 0.096 |
| BB | 0.177 | 0.560 | −0.342 | 0.400 | 0.274 |
| LB | 0.461 | 0.179 | 0.153 | 0.384 | 0.325 |
| RB | 0.316 | 0.372 | −0.164 | 0.454 | 0.068 |
| TC | 0.418 | −0.085 | 0.151 | −0.176 | 0.553 |
| BC | 0.438 | −0.169 | −0.234 | −0.084 | 0.507 |
| LC | 0.243 | 0.059 | 0.829 | −0.280 | 0.485 |
| RC | −0.185 | 0.666 | 0.168 | −0.406 | 0.082 |
| Variance contribution(%) | 49.691 | 21.783 | 13.185 | 59.186 | 26.060 |
| Cumulative contribution to variance(%) | 49.691 | 71.474 | 84.659 | 59.186 | 85.246 |

Notes.

PC1 denotes the first principal component, PC2 denotes the second principal component, PC3 denotes the third principal component, TB, BB, LB, RB, TC, BC, LC and RC denote trunk biomass, leaf biomass, branch biomass, root biomass, trunk carbon content, leaf carbon content, branch carbon content and root carbon content, respectively.

that the eigenvalues of the first three principal components of Scots pine near-mature forests were greater than 1, which could explain 49.691%, 21.783%, and 13.185% of the variance in variance, respectively, and the cumulative variance contribution ratio reached 84.659%, which exceeded 80%, and contained most of the information of the original variables, therefore, the first three principal components were used as the basis for the comprehensive evaluation of Scots pine near-mature forests. The first principal component was mainly loaded by TB, LB, TC and BC, with values of 0.443, 0.461, 0.418 and 0.438, respectively. The second principal component was mainly loaded by BB, RB and RC, and the third principal component was mainly loaded by LB, LC and RC (Fig. S1, Table 2). The screened principal components were comprehensively analyzed to construct the composite principal component PC = 0.58PC1 + 0.257PC2 + 0.156PC3, and the mean of the three replicates was used as the score of the principal components for the comprehensive ranking. The results showed that the comprehensive rankings of the three densities in the near-mature forest were 3, 2, and 1, respectively (Table 3), so the low density of the near-mature forest of Scots pine had the best carbon sequestration capacity, and the high density was the worst.

The eigenvalues of the first two principal components of the mature forest of Scots pine were greater than 1, which could explain 59.186% and 26.060% of the variance of the variance, respectively, and the cumulative variance contribution rate reached 85.246%, which was more than 85%, and it could retain the original information of the variables very well, so only the first two principal components were used as the basis for the comprehensive evaluation of the mature forest of Scots pine. The first principal component was mainly loaded by TB, BB, LB and RB, with values of 0.452, 0.400, 0.384 and 0.454, respectively. The second principal component was mainly loaded by TC, BC and LC (Fig. S2, Table 2). The composite principal component PC=0.694PC1+0.306PC2 was constructed. The

**Table 3  Comprehensive score of carbon sequestration ability of Scots pine with different densities.**

| Number | PC1 | | PC2 | | PC3 | PC | | Ranking | |
|---|---|---|---|---|---|---|---|---|---|
| | NMS | MS | NMS | MS | NMS | NMS | MS | NMS | MS |
| HD | −1.181 | −1.215 | 0.253 | 0.500 | 0.450 | −0.550 | −0.690 | 3 | 3 |
| MD | 0.423 | 0.168 | −1.217 | −1.185 | −0.224 | −0.102 | −0.246 | 2 | 2 |
| LD | 0.759 | 1.047 | 0.964 | 0.685 | −0.226 | 0.653 | 0.936 | 1 | 1 |

Notes.
  PC1 denotes the first principal component, PC2 denotes the second principal component, PC3 denotes the third principal component, PC denotes the composite principal component score, NMS and MS denote near-maturity and mature forests, respectively, and HD, MD, and LD denote high-density, medium-density, and low-density, respectively.

results showed that the three densities of mature forests had a composite ranking of 3, 2 and 1, respectively (Table 3). Taken together, the mature forests of Scots pine also had the best carbon sequestration capacity at low densities, followed by medium densities, and the worst at high densities.

### Carbon stocks of Scots pine plantation forests in the sandy areas of western Liaoning Province, China

The results of the distribution map of Scots pine plantation forests in the Liaoning Sandland showed (in 2023) that the area of Scots pine plantation forests in the Liaoning Sandland was 11,906.8 hm$^2$, which was mainly distributed in Fuxin City and Chaoyang City (Fig. 8). The area of Scots pine plantation forest in Fuxin City is 7,146.3 hm$^2$, with the largest area and carbon stock of MS; the area of Scots pine plantation forest in Chaoyang City is 4,020.4 hm$^2$, with the largest area and carbon stock of NMS and the smallest amount of OMS; the area of Scots pine plantation forest in Jinzhou City is 308.4 hm$^2$, with the largest area of YS, and the largest amount of HMS, with the largest amount of carbon stock, at 2,007.65t; and the area of Scots pine plantation forest in Huludao City is 4,906.8 hm$^2$. The area of Scots pine plantation forest was 471.7 hm$^2$, and the carbon stock of NMS was the largest, 38,549.46t, and there was no distribution of YS and OMS. All of the Scots pine plantation forests were dominated by NMS, NMS and MS, and the number and carbon stock of YS and OMS were small (Table 4, Fig. S3). The uneven distribution of Scots pine plantation forests in the sandy areas of western Liaoning is not only limited to the number of different forest ages but also to the spatial configuration, which will inevitably affect the effectiveness of windbreaks and sand fixation in some areas.

## DISCUSSION

### Analysis of biomass and carbon content in different forest ages

The biomass allocation of trees to each organ shows their survival strategy (*Puglielli et al., 2021*). Changes in the biomass of each organ in Scots pine at different stand ages show that the biomass of each organ in Scots pine grows with the age of the stand, but the rate of growth varies. were slower, and each organ grew at different rates. The biomass growth of each organ in Scots pine was constrained by the size of the individual, indicating that Scots pine biomass allocation is consistent with the theory of allometry growth, which supports our first hypothesis and is also consistent with the findings of *Forrester et al. (2017)* and *Poorter et al. (2015)*. Scots pine allocates biomass mainly to branches and leaves during the

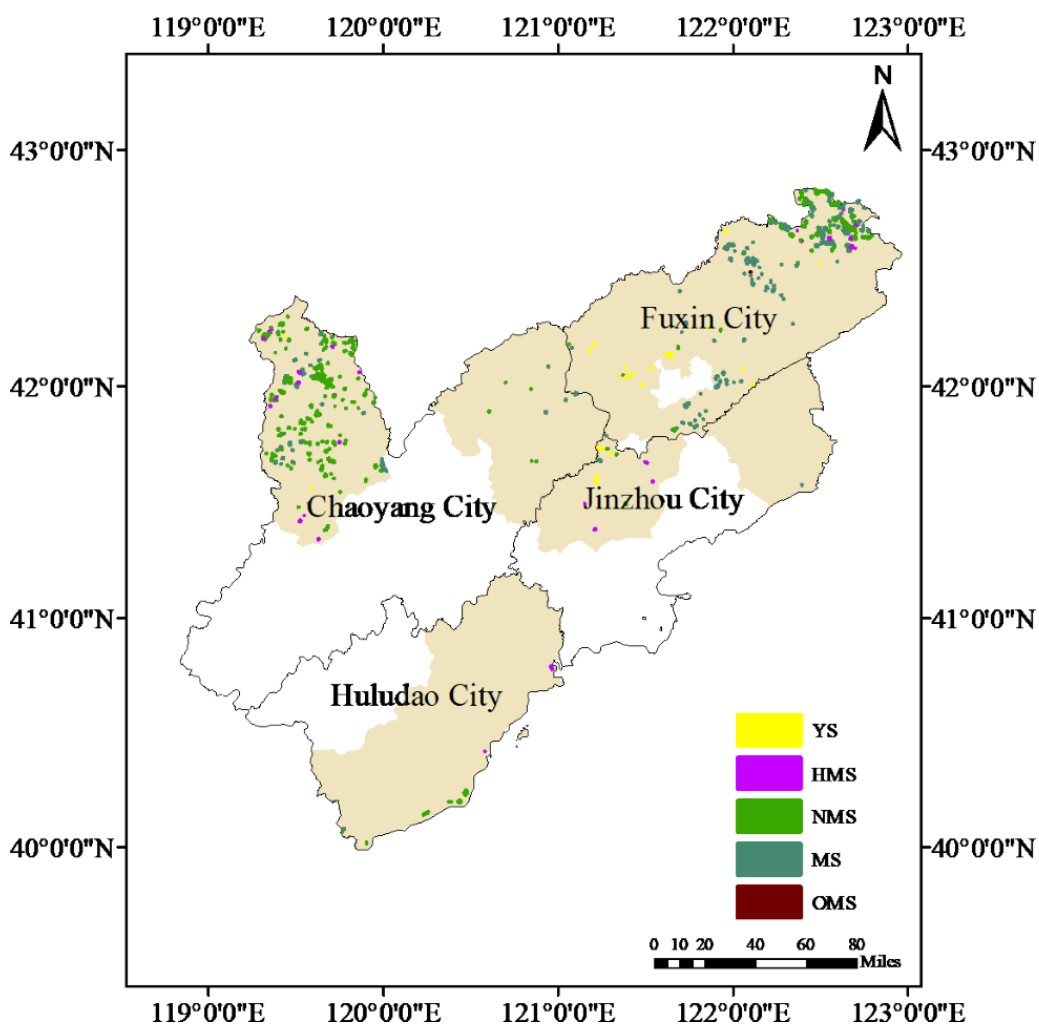

**Figure 8   Distribution of Scots pine plantation forests of various ages in the serious land sanding areas in western Liaoning Province.** The yellow-brown color in the figure indicates the serious land sanding area, and YS, HMS, NMS, MS and OMS indicate the young, half-mature, near-mature, mature and over-mature forests, respectively.

YS and HMS forest periods, *i.e.,* when it is young, whereas in the middle-aged and old-aged periods of NMS, MS, and OMS, more biomass is allocated to trunks and roots, and the plant's foliage can store light energy as carbohydrates through photosynthesis (*Therby-Vale et al., 2022*), suggesting also that, in semi-arid sandy sites Scots pine allocates more biomass to leaves and branches to support faster growth rates in younger ages, and more biomass to trunks and roots in middle and older ages to increase the chances of survival in the local semi-arid as well as sandy and windy environments. This also suggests that Scots pine preferentially allocates biomass to trunk and roots when exposed to drought and wind-sand stress, in line with the optimal allocation theory (*McCarthy & Enquist, 2007*), and supports the first hypothesis.

**Table 4  Carbon stock of Scots pine plantation forests in western Liaoning Sandy Land in 2023.**

| City | Age class | Total area (hm²) | Total number of plants (10⁴) | carbon stock (10 t) |
|------|-----------|------------------|------------------------------|---------------------|
| | YS | 230.1 | 35.439 | 233.224 |
| | HMS | 33.5 | 6.395 | 200.765 |
| Jinzhou | NMS | 11.2 | 1.165 | 75.273 |
| | MS | 33.6 | 1.928 | 129.191 |
| | OMS | 0 | 0 | 0 |
| | YS | 278.7 | 26.839 | 176.627 |
| | HMS | 313.6 | 33.986 | 1066.956 |
| Fuxin | NMS | 2,808.1 | 286.796 | 18,530.463 |
| | MS | 3,733.2 | 340.535 | 22,818.569 |
| | OMS | 12.7 | 2.271 | 305.431 |
| | YS | 15.1 | 3.491 | 22.974 |
| | HMS | 270.8 | 34.941 | 1,096.938 |
| Chaoyang | NMS | 3,063.3 | 478.398 | 30,910.251 |
| | MS | 670.7 | 87.566 | 5,867.622 |
| | OMS | 0.5 | 0.031 | 4.169 |
| | YS | 0 | 0 | 0 |
| | HMS | 23.8 | 4.683 | 147.018 |
| Huludao | NMS | 402.4 | 59.663 | 3,854.946 |
| | MS | 5.5 | 1.204 | 80.678 |
| | OMS | 0 | 0 | 0 |

**Notes.**

YS, HMS, NMS, MS and OMS denote young, half-mature, near-mature, mature and over-mature forests, respectively.

Carbon is a fundamental building block for plant growth and metabolism, and carbon is used to build plant body tissues and cellular structures. At the same time, high carbon content was not exhibited in organs with high biomass, suggesting that the high amount of dry matter in Scots pine may not be determined by carbon content. The carbon contents of organs other than branches of Scots pine roughly showed that they increased firstly and then decreased with the increase of forest age, which was consistent with the study of carbon contents of leaves of oil pine plantation forests by *Wang et al. (2022)*, and also supported the part of the second hypothesis that the carbon contents of organs of Scots pine differed at different forest ages. The peculiar variation in branch carbon content may be related to the aging of branches, and during the field investigation and sampling, we found that young forest-aged branches of Scots pine were young and vigorous, while the number of dead branches in mature forests was high. However, further research is needed to investigate how changes in carbon content affect plant growth, development and physiological functions.

In the course of the study, we found that the rate of increase in biomass per plant became faster in the mature to over-mature forest stage, which may be because the over-mature forest of Scots pine was less dense than the other four forest ages in this experiment. In our management of plantation forests, before the stands reached the over-mature forests,

they had been managed with various kinds of care, and the densities of the stands were all low, and it was difficult to find stands with densities consistent with those of the young and middle-aged forests, which also illustrates the great influence of density on the growth of the trees (*Li et al., 2024*; *Pretzsch, Hilmers & del Rio, 2024*), and also one of the reasons why we studied the effect of density on the growth of Scots pine.

## Analysis of biomass and carbon content at different densities

Density is one of the important factors affecting plant growth (*Pretzsch & del Rio, 2020*; *Thurm & Pretzsch, 2021*). The biomass of each organ and whole plant decreased with the increase of density in both near-mature and mature forests, indicating that low density favors the accumulation of biomass and that the biomass of each organ is different in different densities, which supports our hypothesis II. However, the carbon content of the trunk and root of the near-mature forests of Scots pine did not differ significantly between densities, and this result does not support our assertion that there are differences in carbon content in different densities, which is put forward in our hypothesis II. The carbon content of a single plant of Scots pine ranged from 480 to 620 g/kg, which is similar to the study of *Joosten & Schulte (2002)*. Scots pine root carbon content behaved the same in the three densities of near-mature and mature forests, but other organs behaved differently. The carbon content of each organ was highest at low density in the near-mature forest, but highest at high density in the mature forest, which indicated that the effect of density was different for different stand ages and that the effects on the above-ground and below-ground parts were not the same, which might be because the sampling time of each organ in the near-mature forest of Scots pine was in June, when Scots pine's organs were growing vigorously, and the accumulation of organic carbon was greater than its consumption, so it could accumulate more carbon at the low density. While the sampling time of mature forest was in October when the growth was weakened and the accumulation of organic carbon was less than the consumption because the organs of the low-density were bulkier and the organic carbon consumption was more, which instead made the mature forest have higher carbon content at the high density (*Fang et al., 2016*; *Zhang et al., 2014*). Some studies have shown that plants with high carbon content have higher growth rates and productivity, and can absorb light energy and nutrients more efficiently, thus competing better for resources and having better environmental adaptability (*Liu, Wang & Liu, 2012*; *Ming et al., 2014*), which may also be why the roots and leaves of Scots pine have higher carbon content at high densities. The plant needs to obtain the substances needed for growth through roots and leaves, and in the face of density pressure, to compete for limited resources, the plant will increase the productivity of roots and leaves, which is also in line with the optimal allocation theory. Our study also found that density had a small effect on the carbon content of each organ in the near-mature forest of Scots pine, however, it significantly affected the carbon content of each organ in the mature forest, which may indicate that the survival of the near-mature forest is less restricted by the density condition, which also suggests that we should give full consideration to the relationship between density and growth of the forest stand in the process of operating plantation forests (*Burkhart, 2013*; *Li et al., 2024*). Changes in carbon content show up

differently at different densities in near-mature and mature forests, which could be a form of redistribution. At the same time, it implies that low densities can lift either nutrient availability or light availability, but no doubt the degree of lifting of restrictions varies at different ages.

Here we set only three stand densities in the near-mature and mature forests of Scots pine, and used these three stand densities to correspond to the high, medium and low densities, which means that the high, medium and low densities we set are only relative, and only give the approximate density ranges in which Sphagnum grows better in the near-mature forests and the mature forests, and the optimal density ranges of Scots pine need to be studied more carefully.

### Comprehensive evaluation analysis of the carbon sequestration capacity of Scots pine

When plantation forests are constructed, people always want to cultivate tree species with high carbon sequestration capacity to have higher carbon stocks and healthier forest stands. *Xu et al. (2024)* found that tree size and density were important factors influencing the carbon sequestration capacity of four common tree species in Shanghai. In our comprehensive evaluation of the carbon sequestration capacity of Scots pine at different densities, we found that the carbon sequestration capacity of Scots pine was different at different densities, which supported the third hypothesis. Both near-mature and mature forests of Scots pine showed better carbon sequestration capacity at low densities, and the carbon sequestration capacity gradually decreased with increasing density, which is consistent with the results of *Xu et al. (2024)*. As plants grow, they need more and more space, and different densities lead to different degrees of interspecific competition (*Zhang et al., 2021*). Higher densities imply more intense interspecific competition, especially for large trees such as the near-mature forests and mature forests of Scots pine, which have a higher space requirement, and the higher densities result in limited nutrients and space. The high density makes the nutrients and space limited, which leads to the poor growth of Scots pine, thus reducing its carbon sequestration capacity. Besides, the biomass of forest stands with high carbon sequestration capacity is also higher, which may mean that plants with high carbon sequestration capacity have better growth conditions. Therefore, when plantation forests are built, we should give more consideration to improving the carbon sequestration of the tree species to make them healthier, instead of pursuing the high carbon density, especially in the semi-arid sandy land with poor environmental conditions, and give priority to improving the viability of the plants. In the past, we usually considered the net productivity of trees to express the carbon sequestration capacity of plants (*Zhang et al., 2024*; *Zhou et al., 2021a*; *Zhou et al., 2021b*), but this ignores the influence of plant organs on the carbon sequestration capacity of plants. In this study, eight indicators of biomass and carbon content of four organs of Scots pine were comprehensively considered to measure the carbon sequestration capacity of Scots pine, which is more reasonable. Of course, this study only confirms that in the three densities we set, the lower density in the nearly mature forest of Scots pine shows high carbon sequestration capacity, which

does not mean that it is the density of Scots pine that has the highest carbon sequestration capacity, and more accurate results need to be divided into smaller density gradients.

The findings of this study are important for future silviculture. Perhaps we can mathematically determine the turning point in the accumulation of biomass and carbon sequestration in plantation forests, which can inform decisions about turnover of plantation forests. Similarly, perhaps we can establish a mixed-age forest that achieves a balance between its annual carbon sequestration and biomass growth.

## CONCLUSION

The biomass of each organ of Scots pine in Liaoning sandland increased with the increase of forest age, and the biomass distribution of each organ followed the theory of allometry growth and optimal distribution. The carbon content of Scots pine in different forest ages firstly increased and then decreased, and reached the highest in the near-mature forest. The biomass of each organ in the near-mature and mature forests of Scots pine was constrained by the density, and the biomass decreased with the increase in density. Density has no significant effect on the carbon content of each organ in the above-ground part of Scots pine, but the carbon content of the root system increases with the increase of density. The carbon sequestration capacity of Scots pine plantation forests decreased with the increase in density. In this study, the spatial distribution pattern of carbon stock in Scots pine plantation forests in the west Liaoning sandy land was investigated, and the carbon stock in Scots pine plantation forests in the west Liaoning sandy land was estimated for the first time. Therefore, the effects of forest age and density on biomass and carbon content should be taken into account when plantation forests are established in the west Liaoning sandy area, to give full play to the multifunctional benefits of plantation forests.

## ACKNOWLEDGEMENTS

We are grateful for the theoretical guidance from the Liaoning Provincial Academy of Agricultural Sciences, and the research site provided by the Liaoning Institute of Sand Management and Utilization.

### Funding

This work was supported by the Liaoning Provincial Science and Technology Department (grant numbers 2021JH2/10200007). The funders had no role in study design, data collection and analysis, decision to publish, or preparation of the manuscript.

### Grant Disclosures

The following grant information was disclosed by the authors:
The Liaoning Provincial Science and Technology Department: 2021JH2/10200007.

### Competing Interests

The authors declare there are no competing interests.

## Author Contributions

- Xin Ai conceived and designed the experiments, performed the experiments, analyzed the data, prepared figures and/or tables, authored or reviewed drafts of the article, and approved the final draft.
- Xiangyu Yang conceived and designed the experiments, performed the experiments, analyzed the data, authored or reviewed drafts of the article, and approved the final draft.
- Zhaowei Zhang conceived and designed the experiments, analyzed the data, authored or reviewed drafts of the article, and approved the final draft.
- Hao Chen conceived and designed the experiments, analyzed the data, authored or reviewed drafts of the article, and approved the final draft.
- Wenhui Tang conceived and designed the experiments, performed the experiments, authored or reviewed drafts of the article, and approved the final draft.
- Qingyu Wang conceived and designed the experiments, analyzed the data, authored or reviewed drafts of the article, and approved the final draft.
- Yutao Wang conceived and designed the experiments, performed the experiments, analyzed the data, prepared figures and/or tables, authored or reviewed drafts of the article, and approved the final draft.
- Ping Liu conceived and designed the experiments, performed the experiments, analyzed the data, prepared figures and/or tables, authored or reviewed drafts of the article, and approved the final draft.

## Data Availability

The raw measurements are available in the Supplemental File.

## Supplemental Information

Supplemental information for this article can be found online at http://dx.doi.org/10.7717/peerj.19232#supplemental-information.

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
