# Peer review of "Biomass allocation, carbon content change and carbon stock distribution of Scots pine (Pinus sylvestris var. mongholica) plantation forests at different stand ages and densities in the sandy area of western Liaoning Province, China"

_PeerJ, doi:10.7717/peerj.19232_

## Round 0.1 · original submission · Minor Revisions

A well-written article with robustness in the methods and analysis.

Please make necessary revisions according to both reviewers and we are happy to accept the article

Reviewer 1 ·

Basic reporting

The article is very well prepared and contains the results studied in detail. Afforestation efforts and their success, especially in stressful areas, are of vital importance. Examining both the success and carbon sequestration potential of such afforestation will shed light on the applications. The article contains important results in these aspects and can be recommended for acceptance.

The tree specifically studied in the article is a variety and there are some errors in its Latin spelling. It will be sufficient to write the Latin name once and correctly. Its English name is stated as "Pine". Instead of this common name, the common name of the specific variety must be written. Throughout the text, it would be appropriate to continue with the specific common name only after specifying the common name (Latin name) in the Abstract section.

Experimental design

Throughout the article, there are references such as "Chen Zheng et al. showed that the highest carbon stocks 79 are expected in northeastern China by 2060 (Chen et al., 2024)." Instead of writing the references in this way, it would be more appropriate to write them as "Chen Zheng et al. (2024) showed that the highest carbon stocks are expected in northeastern China."

Research question is well defined, relevant & meaningful. However, the aim of the study may clearly been writen just after the hypotheses.

Validity of the findings

The results and analysis looks meaningful and include important comparisons from different age classes and land features.

Additional comments

This paper may be accepted after making some minor corrections.

Reviewer 2 ·

Basic reporting

The article is mostly well-written and covers an interesting, relevant and timely topic - studying biomass accumulation and carbon content of pine trees in sandy areas in different ages and stand densities.
Some parts are slightly unclear and could be worked on:
line 34: not clear what the forest age means here because it is just introduced later
41-43: this (very important) sentence could be written to be more clear and precise. What does "reasonable" mean in this context? You have great results, I am sure you can make a stronger last sentence.
85: I don't think all methods referred to are covered by the reference given
Overall some sentences could be made a bit shorter to make it easier to understand but it is already quite good as it is.

Experimental design

The research area and sample forests are well chosen and the methods are suitable for the research. The research question is clear and relevant and addresses a knowledge gap that can inform forestry decisions in the future. The methods are described well and experiments could be repeated based on the information given.

Validity of the findings

The manuscript is underplaying the importance of the results for future forestation decisions. Knowing when the turning point in carbon sequestration/biomass accumulation is in time (age) and planting can inform decisions on new plantations and plantation turnover. It should be possible to mathematically determine this point. The authors could discuss a strategy. Is it difficult to establish a new forest and is this part of the decision when to remove a plantation and start anew? Is a mixed age forest a compromise?
I do miss a bit of a discussion about carbon sequestration per area in the manuscript. It is focused on the tree but obviously different densities of planting will affect the sequestration per area which can affect land use decisions and the timing of plantation turnover.
I was also really interested in the changes in carbon allocation. In near mature forests, there is much more biomass in the roots in low stand density but in mature forests this shifts. C content decreased (maybe re-allocated?) and the biomass differences mostly occur in the trunk and branches. I was also wondering what the limitations would be that are relieved by the lower density. Is it nutrient availability or light availability?

Additional comments

Some small comments about the figures (that are all very nice!):
Fig7: make (A) and (B) more recognizable - bold and bigger
Fig8,9,10: I have not tested it but red and green colours are notoriously difficult to separate for colour blind people, so if you could test this (if you have not already done) and adjust if necessary, that would be great.

---

## Round 0.2 · accepted · Accept

Please change Scotch Pine to Scots Pine throughout the whole manuscript as suggested, and congratulations.

Reviewer 2 ·

Basic reporting

I am happy with the publication now - all of my concerns have been addressed. Please change Scotch Pine to Scots Pine though throughout the whole manuscript.

Experimental design

See above. All concerns have been addressed.

Validity of the findings

See above. All concerns have been addressed.